# Nucleic Acid Detection with Ion Concentration Polarization Microfluidic Chip for Reduced Cycle Numbers of Polymerase Chain Reaction

**DOI:** 10.3390/mi13091394

**Published:** 2022-08-26

**Authors:** Chengzhuang Yu, Shijie Dai, Shanshan Li, Junwei Li, Hezhi Hu, Jiyu Meng, Chunyang Wei, Jie Jayne Wu

**Affiliations:** 1Hebei Key Laboratory of Smart Sensing and Human–Robot Interactions, School of Mechanical Engineering, Hebei University of Technology, Tianjin 300130, China; 2State Key Laboratory of Reliability and Intelligence of Electrical Equipment, School of Electrical Engineering, Hebei University of Technology, Tianjin 300130, China; 3Institute of Biophysics, School of Health Sciences and Biomedical Engineering, Hebei University of Technology, Tianjin 300130, China; 4Department of Electronics and Information Engineering, Hebei University of Technology, Langfang 065099, China; 5Department of Electrical Engineering and Computer Science, The University of Tennessee, Knoxville, TN 37996, USA

**Keywords:** microfluidic, nucleic acid detection, polymerase chain reaction, electrokinetic preconcentration, ion concentration polarization

## Abstract

Nucleic acid detection is widely used in disease diagnosis, food safety, environmental monitoring and many other research fields. The continuous development of rapid and sensitive new methods to detective nucleic acid is very important for practical application. In this study, we developed a rapid nucleic-acid detection method using polymerase chain reaction (PCR) combined with electrokinetic preconcentration based on ion concentration polarization (ICP). Using a Nafion film, the proposed ICP microfluidic chip is utilized to enrich the nucleic acid molecules amplified by PCR thermal cycles. To demonstrate the capability of the microfluidic device and the hybrid nucleic-acid detection method, we present an animal-derived component detection experiment for meat product identification applications. With the reduced cycle numbers of 24 cycles, the detection can be completed in about 35 min. The experimental results show that this work can provide a microfluidic device and straightforward method for rapid detection of nucleic acids with reduced cycle numbers.

## 1. Introduction

As the basic building blocks of living organisms, nucleic acids are ubiquitously present in animal and plant cells, viruses and bacteria. The identification of nucleic acid molecules for biotechnology and medical diagnostics has been a subject of intense studies in the past decades [1,2]. Continuous development of rapid and sensitive methods to detect specific nucleic acids, such as point-of-care testing (POCT) [3,4], is very important for practical applications, e.g., pathogen detection [5], and general laboratory tasks. A traditional nucleic acid detection technique is through molecular hybridization with deoxyribonucleic acid (DNA) probes [6,7]. It works by immobilizing a single-stranded DNA sequence on a surface as the recognition element, and the target single-stranded DNA is recognized by its specific binding affinity to the complementary nucleotide sequence. The hybridized DNA molecules can be detected by various labeling strategies, such as enzymes, chemical fluorescence, isotope or biotin labeling methods [8]. Currently, amplification of DNA samples at low concentrations by polymerase chain reaction (PCR) combined with electrophoresis or fluorescent labeling as detection methods have been routinely used for genetic testing [9,10].

The PCR method is a well-established targeted technology, including gel electrophoresis PCR [11], real-time fluorescence quantitative PCR (qPCR) [12,13,14], and digital PCR (dPCR) [15,16]. While instruments based on PCR are commercially available, the main disadvantage of this method is the long experimental operation time. The reason is that the target nucleic acid molecules need to be amplified sufficiently to ensure detection, and the amplification process usually takes 35~40 reaction cycles. Some efforts have been taken to reduce the cycling time. However, it still needs no less than 40~60 min to obtain a detectable nucleic acid concentration. Such a long time is not suitable for some situations where a rapid detection is required. Furthermore, more cycles require a longer operation time, and also lead to more accumulation of point mutation artifacts [17]. Thus, it is beneficial to minimize PCR cycles.

Unfortunately, reducing cycles may not be able to enrich the target nuclei acids adequately. One possible solution is to include an extra preconcentration step for targeting nucleic acids. In our previous work, we have developed several kinds of surface-based affinity biosensors [18,19,20,21,22,23] to preconcentrate DNA, IgG, metal ions or other targets with the help of ACEK (Alternating Current Electrokinetics) effects. However, surface-based biosensors require good quality surface functionalization, which is challenging. Up to now, researchers have proposed many methods to preconcentrate analytes, such as dielectrophoresis (DEP) [24], alternating current electro-osmosis (ACEO) [25], field amplified sample stacking (FASS) [26], isoelectric focusing (IEF) [27], isotachophoresis (ITP) [28] and ion concentration polarization (ICP) [29], etc. Among these methods, DEP and ACEO are susceptible to electrothermal flow due to the high electrical conductivity of the PCR reaction system. FASS is a type of capillary electrophoresis technique. It usually requires a buffer with higher conductivity than the sample solution to form the enrichment interface. Therefore, it is not suitable for the enrichment of PCR reaction systems. IEF and ITP require complex buffers and experimental setups and are mostly used for the separation of different ions and proteins.

Compared with these methods, the reaction system of ICP has many compatible with that of PCR. ICP has great advantages in the preconcentration of target molecules due to its easy implementation and high efficiency [30]. This phenomenon is caused by the selective transportation of ions in the electrolyte near a nano-porous membrane [31]. The low-abundance PCR products can be concentrated by ICP to increase the local concentration in a short time. In a typical PCR reaction system, the reaction solution contains a large amount of free Mg^2+^, even though they are consumed during the amplification process. So, it is possible to enrich DNA molecules by ICP because they carry negative charges in the PCR reaction solution. Therefore, PCR technology can be combined with ICP, PCR reaction for amplification, and then ICP to concentrate DNA molecules with fluorescent labels, which can shorten amplification time and improve detection efficiency. Although detection of a reduced cycle numbers of PCR has been achieved using methods such as genotyping [32] and electrochemistry [33], to our knowledge, the work is the first to report the realization of a rapid sample-to-answer nucleic acid detection with a PCR-ICP hybrid microfluidic device.

To overcome the limitation of the long operation time, here in this work, we developed a rapid nucleic-acid detection method using PCR for amplification and ICP for preconcentration. In our workflow, nucleic acid amplification needs only 24 cycles, then the amplified nucleic acid was concentrated by the ICP effects. With fewer cycles (24 cycles in this work, and 35~40 cycles in state-of-art works), the possibility of point mutation artifacts is significantly reduced. Specifically, both the PCR and ICP operations are integrated into one microfluidic chip. Finally, we validate and demonstrate the usefulness of our hybrid nucleic-acid detection method with an application of animal-derived ingredient detections.

## 2. Materials and Methods

### 2.1. Working Principles

#### 2.1.1. Ion Concentration Polarization

The working principle of ion concentration polarization is shown in Figure 1A. The microchannel is divided into an anode region and a cathode region by a Nafion membrane. When a DC voltage is applied across the microchannel, the cations near the membrane migrate through the Nafion membrane due to the high difference in conductivity between the Nafion film and the solution. The anions on the anode side move away from the membrane and generate an ion-depletion zone. Charged biomolecules are trapped and accumulated on the edge of the depletion zone due to a force balance between electro-osmotic flow and electrophoresis, which is induced by an electric field gradient in the main channel.

#### 2.1.2. Cycling Probe Method for Fluorescence Analysis

To generate a distinct fluorescent signal, fluorescence-labeled based cycling probe method [34] was used in this paper. Carboxy fluorescein (FAM, ex/em 492/518 nm, CAS: 3301-79-9, Aladdin, Shanghai, China) was used as the fluorescent label herein. Labeling nucleic acids with fluorescent groups is a common approach to detecting nucleic acids. Using the fluorescent properties of the labels, the target signal could be detected by the cycling probe method. The cycling probe method is a high-sensitivity detection method using a hybrid probe composed of RNA and DNA combined with endonuclease RNase H. It can efficiently detect the target gene fragments during the amplification process.

As shown in Figure 1B, the cycling probe contains the RNA bases, the fluorescent reporter group, and the fluorescent quenching group. Among them, the RNA bases are free-state distributed in the solution. The fluorescence of the reporter group is inhibited due to fluorescence quenching when the probe is complete with the reporters. However, when the probe is hybridized with the complementary sequence in the amplification product, RNase H cuts off the probe in the RNA part, the quenching inhibition is relieved, and the reporter group emits fluorescence. Thus, the amplified nucleic acid product can be detected by measuring the fluorescence intensity.

### 2.2. Fabrication of the Microfluidic Device

As shown in Figure 2(Ai), a polydimethylsiloxane (PDMS, Dow Corning Sylgard 184, Midland, MI, USA) microfluidic chip replica (50 μm in depth, 200 μm in width, and 1 mm in length), indicated by “PDMS *”, was used to hold the Nafion solution and to form a thin Nafion film. Another PDMS replica, indicated by “PDMS #” (30 μm in depth, 100 μm in width, and 40 mm in length), was used to hold the nucleic acid samples. Both of them were fabricated by soft lithography.

The standard procedure for soft lithography was as follows. First, an on-demand pattern was drawn using AutoCAD and printed onto a transparent film by a high-resolution printer. Then the pattern was transferred onto a silicon wafer which was coated with SU8 photoresist (MicroChem, Westborough, MA, USA). Then the SU8 master mold was subsequently treated with dimethyldichlorosilane (Sigma-Aldrich, St. Louis, MO, USA) in a vacuum drying oven for 12 h at room temperature. Next, PDMS prepolymers (PDMS: curing agent = 10:1 *w*/*w*) were poured onto the master mold. After being cured at 80 °C for 2 h, the PDMS was peeled off from the master.

After punching two through-holes as the inlet and outlets, the PDMS block for making ion selective membranes (shown as PDMS *, in Figure 2(Ai)) was held by two magnets to get reversible packaged with a clean glass slider. Then a drop of 1 μL Nafion solution was introduced into the microchannel by the capillary force. After a cleaning step to remove excess Nafion solution and curing at 90 °C for 10 min, a thin Nafion film will form on the glass slider, as shown in Figure 2(Aii). As to the PDMS block for observing experimental results (named PDMS #, Figure 2(Aiii)), it was plasma bonded with a glass slider, as shown in Figure 2(Aiii). The surface of PDMS and glass slides to be bonded were plasma treated for the 40 s in the plasma machine (PDC-002, Harrick, New York, NY, USA).

Figure 2B demonstrates the assembly of the PCR-ICP hybrid microfluidic device. Two silver wire electrodes were inserted into the inlets and outlets. Figure 2C gives a photo image of the assembled ICP microfluidic chip and here also provides the microscopic image of the ICP area.

### 2.3. Equipment and Reagents

To verify the performance of the PCR-ICP hybrid nucleic-acid detection method proposed in this work, we conducted detection experiments on the application of animal-derived ingredients. We used CycleavePCR™ meat species identification kit (Takara, includes enzyme, buffer, dNTP mixture, primer and probe, Dalian, China) and DNA extraction reagent (Takara, MightyPrep reagent for DNA, Dalian, China). A program-controlled thermal circulator made of the thermocouple was used for PCR. For concentration experiments, 20% Nafion (Perfluorosulfonic acid-polytetrafluoroethylene copolymer, DuPont, Wilmington, DE, USA) solution was used to obtain ion selective exchange membrane, polydimethylsiloxane (PDMS, Dow Corning Sylgard 184, Dow Corning Corporation, Midland, MI, USA), and glass slides were used to form the microchannel and substrate. A DC power supply (Maisheng, Shenzhen Mestek Tools Co., Ltd., Shenzhen, China) was used to apply electrical potential. For imaging, an inverted fluorescence microscope (Nikon, Ti-S, Tokyo, Japan) and a CCD camera (Nikon, DS-Qi2, Tokyo, Japan) were used. An ultraviolet (UV) spectrophotometer (UV5-Nano, Mettler Toledo, Zurich, Switzerland) was used to provide quantitative detection of the initial concentration of the targets.

### 2.4. Sample Preparation

The species of meat used in the study were beef and chicken, and were purchased from a supermarket in Tianjin, China. DNA was extracted from the samples and purified using the MightyPrep™ (Takara MightyPrep reagent for DNA, Dalian, China) according to the manufacturer’s protocol. First, we put 25 mg of chopped meat sample into a 1.5 mL tube. Next, we added 100 μL sterilized water into the test tube, covered tightly and kept at 95 °C for 5 min. Finally, we centrifuged at 12,000 rpm for 5 min and took the supernatant as the test sample. A reagent blank was included with each DNA extraction as a negative control. The concentration and purity of the extracted DNA were quantified using the UV spectrophotometer. All DNA samples were stored at −20 °C before use.

### 2.5. Experimental Setup and Data Analysis

The PCR reaction system consisted of the following: 12.5 μL 2 × CycleavePCR Reaction Mixture, 5 μL Primer·Probe Mix-1 (for chicken), 5 μL DNA template (sample, positive control template or blank control), 2.5 μL double-distilled water (ddH_2_O), the total volume of the reaction system is 25 μL. For the negative control, sterile distilled water was used instead of the test samples, and for the positive control, the standard control DNA (2 ng/μL) in the kit was used as template DNA.

The experimental steps for PCR-ICP detection are shown in Figure 2D. First, the reaction components other than the test samples were prepared and sub packed into the reaction tube. Then the test samples, positive control and negative control were added. The above reaction solution must be prepared on ice. Next, the prepared PCR reaction solution was injected into the main channel by a pipette. Two silver electrodes were inserted into the outlet and inlet. The diameter of the silver electrode was slightly larger than that of the inlet and outlet to prevent reagent volatilization. Then, we put the chip into the thermal circulator for PCR amplification. The reactions were performed in our system under the following conditions: 95 °C for 10 s, followed by cycles at 95 °C for 5 s, 57 °C for 10 s, and 72 °C for 20 s. The number of cycles will be discussed in a later section. Finally, a DC electric field is applied by the DC power supply and the fluorescence signal was observed by a fluorescence microscope. For data analysis, Image J (NIH, Bethesda, MD, USA), and Origin 9.1 (OriginLab, Northampton, NC, USA) were used when required.

## 3. Results and Discussions

### 3.1. Preconcentration Performance of the Device

To enable rapid nucleic acid detection in our device, the preconcentration mechanism of ICP is the key. To test the concentration effect of the device, we validated it using different concentrations of sodium fluorescein aqueous solution. Sodium fluorescein is an organic compound, with the molecular formula C_20_H_10_Na_2_O_5_, which exists in an aqueous solution as Na^2+^ and negatively charged fluorescein molecules and emits yellow-green fluorescence (ex/em 490/513 nm). This fluorescence spectrum is similar to that of the experimental DNA fluorescent labels (FAM). The performance of DNA molecules can be simulated with sodium fluorescein, because under physiological conditions, the phosphate groups in the sugar-phosphate backbone of nucleic acid molecules are ionic, and in this sense, DNA polynucleotide chains can be considered polyanionic. When the device was applied with DC power, Na^2+^ passed through the cation-selective permeation membrane (Nafion membrane) and an ion-depletion zone was formed in the anode region. Under the action of electro-osmotic flow and electric field, the negatively charged fluorescein molecules appear enriched outside the ion-depletion zone. As shown in Figure 3A, fluorescence intensity enhancement can be observed after 5 min of applying the electric field. Here an electric field of 25 V/cm was used, and a 400 μm wide Nafion film was patterned on the glass surface.

We also performed enrichment experiments with different concentrations of sodium fluorescein in an aqueous solution. The relative fluorescence intensities of three different concentrations of sodium fluorescein aqueous solutions are shown in Figure 3B. The fluorescence intensity is shown to increase approximately linearly with time at concentrations of 10 ng/μL. At the initial concentration of 10^2^ ng/μL, the fluorescence intensity reaches an inflection point at about 4 min and increases slowly thereafter. This is because the accumulation of DNA molecules reaches a state close to saturation, after which the accumulation becomes slower due to the space limitation. When the initial concentration was 10^3^ ng/μL, faint fluorescence can be observed in the microchannel at t = 0 s, and reaches the upper limit of fluorescence intensity within the next 100 s. After 100 s, the fluorescence intensity no longer changed, probably because the fluorescein molecules that can be carried per unit volume reach an upper limit. Therefore, fluorescein molecules per unit volume no longer increase, and fluorescence intensity remains the same. Then, the accumulation of fluorescein only increases the area of the preconcentration zone and the fluorescence intensity does not increase, as shown in Figure 3C. Figure 3C shows the experimental fluorescence accumulation at the concentration of 10^3^ ng/μL. At t = 0 s, the molecules and ions in the solution in the microchannel were uniformly distributed, and there was a faint fluorescence in the microchannel. During the next 100 s, fluorescein molecules gradually accumulate in the preconcentration zone due to the effect of ICP, and an accumulation of fluorescent signals can be observed. After 100 s, the fluorescein molecules per unit of volume gained saturation. Instead, the area of the preconcentration zone increased. In addition, when starting with 10^2^ ng/μL of sodium fluorescein in the channel, fluorescein molecules could be concentrated and become observable within 5 min.

### 3.2. The Effect of ICP on Preconcentration Zone

According to the experimental results shown in Figure 3, the accumulation of samples in the preconcentration zone is the key to successful detection. It is necessary to analyze the movement of sample molecules in the microchannel. The sample molecules usually experience the following states after the DC power is applied. With the influence of ICP, a large number of cations pass through the Nafion membrane and gradually form the ion-depletion zone. Negatively charged molecules start to accumulate at the edge of the ion-depletion zone, and a gradual increase in fluorescence intensity can be observed. Due to the limitation of ion concentration, electric field and space, the sample molecules reach a limit in the preconcentration zone, after which the fluorescence lightness no longer increases, but shows the expansion of the enriched region. It can be observed in Figure 3B that the initial concentration has an effect on the accumulation rate, thus a suitable initial concentration is also an important parameter for a successful assay.

### 3.3. Application of Meat Products Authenticating

We conducted a rapid detection experiment of the chicken-derived component from meat products to validate our method. To ensure the reliability of the experimental results, a positive experimental group, a negative experimental group, a positive control group (positive template DNA from the kit, for chicken here) and a blank control group (ddH_2_O instead of DNA sample) were set up. DNA samples were extracted from chicken and beef products purchased from a local supermarket, and these samples served as the experimental group. Chicken samples were used as the positive experimental group and beef samples were used as the negative experimental group. Figure 4A shows the experimental results of 20 thermal cycles (about 25 min) followed by 5 min of ICP enrichment at 25 V/cm. No preconcentration zones were generated in the blank control microchannels (Figure 4(Ai)), indicating that there was no chicken-derived DNA in this channel, and also indicating that the experimental reagents were not contaminated. Chicken samples group (Figure 4(Aiii)) showed the same results as the positive template group (Figure 4(Aii)). The preconcentration zones were formed on the anode side of the Nafion membrane, and there were no preconcentration zones in the experimental group of beef samples (Figure 4(Aiv)). The results demonstrate that our device was capable of detecting target genes.

### 3.4. Optimization of Detection Time

By characterizing the preconcentration device in Section 3.1, we found that the fluorescence signal can be enriched and detectable within 5 min when the target molecule concentration was greater than 10^2^ ng/μL. For the purpose of rapid detection, we controlled the enrichment time of ICP to about 5 min. Therefore, it is necessary to amplify the target gene to more than 10^2^ ng/μL. For this purpose, we conducted experiments to test the amplification time of the target gene to 10^2^ ng/μL with different initial concentrations (10^−4^, 10^−3^, 10^−2^, 10^−1^, 10^0^ ng/μL), each group repeated three times, as shown in Figure 4B. The results show that the time required to amplify the target gene to 10^2^ ng/μL was linearly related to the logarithm of the initial concentration. This is because in the initial stage of PCR, the amplification of the target gene exponentially increases with time, as the number of target genes doubles with each thermal cycle. Thus, for every ~3.3 cycles, the number of target genes increase by one order of magnitude. When the target gene was amplified to 10^2^ ng/μL, ICP enrichment was then performed for about 5 min until the fluorescent signal can be detected. In this way, the overall assay time of our device was controlled between 25 and 40 min. To ensure that low concentrations could also be detected, the assay time was set as 30 min (24 thermal cycles) for PCR with 5 min for ICP. Based on the above experimental results, our device can detect the target gene in about 35 min.

## 4. Conclusions and Future Works

Here, we demonstrate a promising microfluidic system, which integrates with PCR and ICP techniques to enable rapid nucleic acid detection. The amplification of nucleic acid was initially performed in the micro channel to obtain a starting concentration for ion concentration polarization. We demonstrated the capability of ICP to identify the fluorescence signals from the products of PCR cycles. With only 24 cycles, the nucleic acid could be amplified into 10^2^ ng/μL and could be further concentrated by the ICP effect. Correspondingly, the total operation time can be reduced to 35 min.

As revealed by the fluorescence images of sodium fluorescein aqueous solution and nucleic acid reagents, the concentration of target DNA was well amplified and concentrated, which is critical to reducing the total operation time. Fewer cycle numbers could decrease the risk of unreliable results caused by the byproducts during PCR cycles. In summary, PCR and ICP techniques are compatible and can work together to realize rapid and reliable nucleic acid detection.

## Figures and Tables

**Figure 1 micromachines-13-01394-f001:**
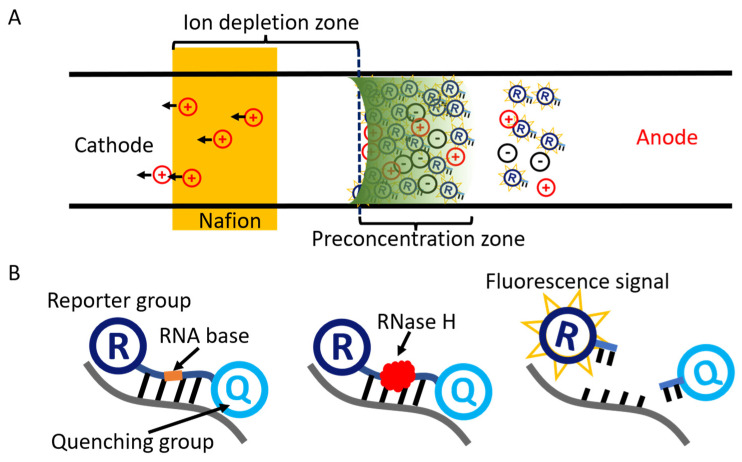
(**A**) Principle of ion concentration polarization. Negatively charged molecules are enriched to the preconcentration zone by the electric field and electro-osmotic flow. (**B**) Schematic of the cycling probe method. The fluorescent reporter group leaves the quenching group in the presence of endonuclease RNase H to emit a fluorescent signal.

**Figure 2 micromachines-13-01394-f002:**
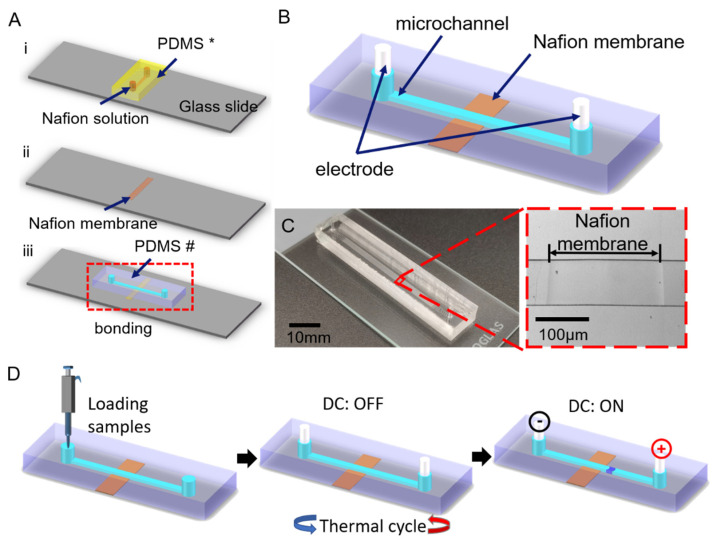
(**A**) Fabrication steps of the device. (**i**) Fabrication of Nafion membrane using PDMS *. (**ii**) Nafion membrane formed on glass substrate. (**iii**) Bonding with PDMS #. (**B**) The specific structure of the PDMS chip. (**C**) The experimental device and the optical micrograph of the Nafion membrane (40 mm length, 100 μm width, and 30 μm depth). (**D**) Steps of PCR-ICP hybrid experimental operation.

**Figure 3 micromachines-13-01394-f003:**
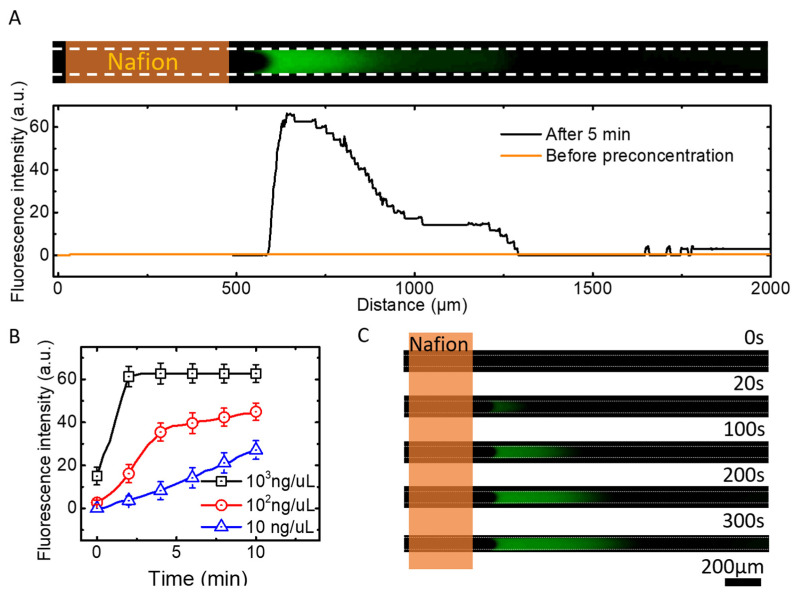
(**A**) Phenomenon of ion concentration polarization with electric field 25 V/cm. (**B**) Fluorescence intensity of different concentrations (10 ng/μL,10^2^ ng/μL and 10^3^ ng/μL) of sodium fluorescein aqueous solution with the preconcentration operation. (**C**) Experimental image of fluorescence intensity changing with time at the concentration of 10^3^ ng/μL.

**Figure 4 micromachines-13-01394-f004:**
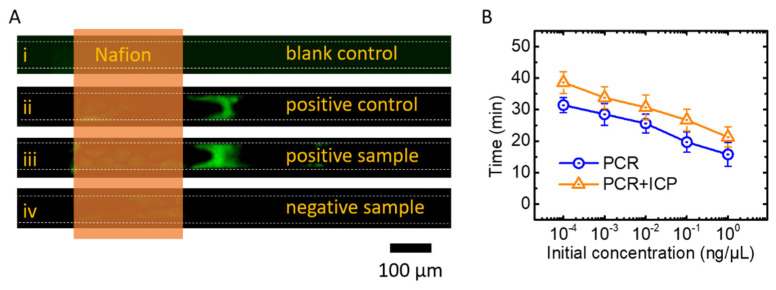
Application of meat products authenticating. (**A**) Results of nucleic acid preconcentration in the experimental and control groups. (**B**) Time as a function of original concentration during nuclei acid amplifications.

## Data Availability

The data that support the findings of this study are available from the corresponding author, upon reasonable request.

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
