# Peer review of "Nucleic Acid Detection with Ion Concentration Polarization Microfluidic Chip for Reduced Cycle Numbers of Polymerase Chain Reaction"

_micromachines, 2022, doi:10.3390/mi13091394_

Round 1

Reviewer 1 Report

Nucleic acid detection with ion concentration polarization microfluidic chip for reduced cycle numbers polymerase -chain reaction cycle numbers

Recommendation: Publish after minor revision

Comments:

The authors present the results of an interesting rapid nucleic acid detection method using polymerase chain reaction (PCR) combined with electrokinetic preconcentration based on ion concentration polarization. This work provides a microfluidic device and rapid detection of nucleic acids with reduced cycle numbers. They claim that the total operation time can be reduced into 35 minutes. While I find their experimental approach is interesting, the methods have lacks of information. Also, I don't feel that they have translated their observations well in the result and discussion part.

Specific comments:

1-Lines between 66-75 are not well structured.

Lines 66-69, Up to now, researchers have proposed many effective methods to  perform a preconcentration, such as dielectrophoresis (DEP) [18], alternating current electroosmosis (ACEO) [19], field amplified sample stacking (FASS)[20], isoelectric focusing (IEF)[21], isotachophoresis (ITP) [22] and ion concentration polarization (ICP) [23] , etc.

Is this merely a demonstration of alternative techniques or an explanation of why you did not use those techniques. What are these techniques' drawbacks?

The authors abruptly altered the subject and mentioned their previous work.

Lines 74, they said the ICP method again. Could you please reorganize lines 66-75 clearly?

2- Line 82, PCR technology can be combined with ICP, i.e., PCR… Why did you use i.e ?

3-Line 138-139 in a vacuum drying oven for 12 hours, what was the temperature?

4-Line 144-145 After a washing step and curing at 90℃ for 10 minutes… Is this washing step for device? membrane? or ..? and with which compound is used?

5-What is the ratio of PDMS and curing agent to make this device?

6-Plasma can change surface properties.  You used plasma bonded with a glass slider (Line 147). Could you explain why you used  plasma bonded with a glass slider, and give more detail about Figure 2 A iii.

7-Figure 2.A Loading sample, how long does it take? How did you provide homogeneity in this channel?

8-How many times did you use this device? Is this disposable after 24 cycles? How did you clean this device? And how were you sure there is no contamination?

9-Figure 2 B. Silver wire electrodes were used. The inlet is tight the high amount of protein can cause problem for liquid flow and it would affect residence time of stream as well. How did you prevent your system these kinds of problems. Please give more detail in material& method about this size of meat, flow etc. Furthermore, what are the reasons of choosing plasma bonded with a glass slider and silver electrode?

10-I have doubt about samples and results, because it says in the sample preparation (2.4) The species of meat used in the study were beef and chicken... and later It says we conducted a rapid detection experiment of chicken-derived component from meat products to validate our method in Section 3.2. The results depend on beef and chicken meat and their amount, ratio is also quite important. But the sections of Sample preparation and results are not well in line. It gives insufficient information about number of meat, chicken sample, experiment time, storing temperature etc.

11-Figure 3.B please give unit for the time.

12-In the result and discussion section, the effect of ICP on preconcentration zone should be given explicitly.

13-The authors say The fluorescence intensity is shown to increase approximately linearly with time at concentrations of 10 ng/μL and 102 ng/μL in lines 229-231. It is not significant to explain this trend as approximately linearly. There are two different trends here.

14-Line 282, our device can detect the target gene in about 35 minutes. How many experiments give this result? What is the reproducibility of this method?

15-Line results caused by the byproducts during PCR cycles, please correct by

Reviewer 2 Report

My comments are listed in the following. 

(1) The authors should avoid using long sentences. The following sentenses need to be reorganized or revised:

a)"The pre-concentration mechanism of ICP is the key to enable our device to achieve 

rapid nucleic acid detection."

b)"we conducted experiments on the application of animal-derived in- 

gredient detections".

c)"Here we extracted DNA samples from chicken and 

beef products purchased from local supermarkets as the experimental group".

d)"indicating 

that there was no chicken-derived DNA in this channel and that the experimental reagents 

were not contaminated."

e) "Here, we have demonstrated a promising ...".

f)"PCR and ICP techniques are complementary to pave the way of rapid and reliable 299

nucleic acid detections"

g)" be optical observed".

h)"more cycles is ..."

i) "amount of free Mg2+, even though it is consumed" should be like " amount of free Mg2+ ions, even though they are consumed ".

(2) No units are shown for the " time " in Fig.3 B .

(3) It looks like ion concentration polarization base microfluidic devices have been designed and implemented in a lot of references .  The authors can check this review article.

" Design and application of ion concentration polarization for preconcentrating charged analytes

Physics of Fluids 33, 051301 (2021); https://doi.org/10.1063/5.0038914".

Then I guess the authors may have to make some statements/discussion about the novelty of this work. The current statement in the introduction part is not enough. Can the authors make the statement that this is the first time that few circles have been achieved by this method of using  ion concentration polarization base microfluidic devices?

(4) The authors didn't use the word " the" in many places through the whole text. Please check and revise accordingly.  
